# Gut Microbiota Influence in Hematological Malignancies: From Genesis to Cure

**DOI:** 10.3390/ijms22031026

**Published:** 2021-01-20

**Authors:** Mireia Uribe-Herranz, Nela Klein-González, Luis Gerardo Rodríguez-Lobato, Manel Juan, Carlos Fernández de Larrea

**Affiliations:** 1Centre de Diagnòstic Biomèdic, Department of Immunology, Hospital Clínic de Barcelona (HCB), 08036 Barcelona, Spain; klein@clinic.cat (N.K.-G.); mjuan@clinic.cat (M.J.); 2Institut D’ Investigacions Biomèdiques August Pi i Sunyer (IDIBAPS), Fundació Clínic per a la Recerca Biomèdica (FCRB), 08036 Barcelona, Spain; lgrodriguez@clinic.cat; 3Plataforma d’Immunoterapia Banc de Sang i Teixits–HSB, Universitat de Barcelona, 08036 Barcelona, Spain; 4Amyloidosis and Myeloma Unit, Department of Hematology, Hospital Clínic of Barcelona, 08036 Barcelona, Spain; 5Plataforma d’Immunoteràpia Hospital Sant Joan de Déu-HCB, Esplugues de Llobregat, 08950 Barcelona, Spain

**Keywords:** microbiota, hematological malignancies, bacterial metabolites, chemotherapy, allogeneic stem cell transplantation, autologous stem cell transplant, CAR-T cell therapy

## Abstract

Hematological malignancies, including multiple myeloma, lymphoma, and leukemia, are a heterogeneous group of neoplasms that affect the blood, bone marrow, and lymph nodes. They originate from uncontrolled growth of hematopoietic and lymphoid cells from different stages in their maturation/differentiation and account for 6.5% of all cancers around the world. During the last decade, it has been proven that the gut microbiota, more specifically the gastrointestinal commensal bacteria, is implicated in the genesis and progression of many diseases. The immune-modulating effects of the human microbiota extend well beyond the gut, mostly through the small molecules they produce. This review aims to summarize the current knowledge of the role of the microbiota in modulating the immune system, its role in hematological malignancies, and its influence on different therapies for these diseases, including autologous and allogeneic stem cell transplantation, chemotherapy, and chimeric antigen receptor T cells.

## 1. Introduction

The complex ecosystem formed by trillions of microorganisms living in and on the human body is known as the human microbiota. The microbial communities inhabiting inside us include bacteria, virus, fungi, yeast, and protozoa. Most of the human microbiota resides in the gastrointestinal tract, but many other microbial niches exist such as the skin, the lungs, the vagina, or the mammary glands. The gastrointestinal tract, with a projected area of 32 m^2^ [1], is among the largest interaction surfaces between the host, environmental factors, and antigens in the human body. The latest study sets an estimation of over 40 trillion intestinal microorganisms, bringing the ratio closer to 1:1 to somatic cells, expected to be around 30 trillion [2]. The diversity of the bacterial community that resides inside the mammalian intestine is influenced by several factors such as physiological features, chemical and nutrient gradients, and the strictly compartmentalized host immune response [3].

The bacteria that comprise the mammal gut microbiota belong primarily to four phyla: *Firmicutes, Bacteroidetes, Proteobacteria* and *Actinobacteria*. Altogether, these phyla account for over 95% of the total bacteria in the mammalian microbiota, regardless of the animal species [4]. The genetic potential of this biomass was estimated in 3.3 million non-redundant microbial genes by the ‘Metagenomics of the Human Intestinal Tract’ project. According to this study, 99.1% of the genes are of bacterial origin, followed by archaeal, eukaryotic, and viral origins. The analyzed cohort contained 1150 bacterial species abundant enough to be detected and at least 160 of such bacterial species are estimated to be in every single person [5].

The gastrointestinal microbiota is generated after birth through vertical transmission and it is then constantly shaped by environmental factors throughout life. The most relevant factors affecting the intestinal microbe composition are: the childbirth delivery method (vaginal vs. cesarean) [6,7,8]; the diet (including breastfeeding as the most significant microbiome variable during the first year of life); household exposures (especially during infancy) [9]; geographical location and genetics. However, the latter has been shown to have a minor role compared to environmental factors [10]. Medication is another important factor that shapes the microbial composition in the human body. Antibiotics are the most relevant drugs affecting gut microbiota composition [11,12], but non-antibiotic drugs like antimetabolites, antipsychotics, and calcium-channel blockers can also affect the human microbiome [13]. Antibiotics reduce the microbial diversity rapidly and significantly; around one third of the total bacterial taxa is lost in 3 days. Four weeks after a course of antibiotics, bacterial communities mostly return to their initial state, even though this return is not complete [12,14]. It has now been established that antibiotics during early life can have a profound effect upon health, increasing the risk of developing asthma [15], type 1 diabetes [16] or having long-term host metabolic effects [17,18]. According to a recent study, the most efficient therapeutic approach to fully recover the initial microbial state is an autologous fecal microbiota transplant (FMT). This approach induced a faster microbial recovery compared to the use of probiotics, which hindered a complete microbiome reconstitution [19].

The gut microbiota and the human host have co-evolved in a mutualistic association [4]. The host obtains benefits like the strengthening of the gut integrity, intestinal epithelium modulation [20], protection against pathogens [21], fat metabolism support [22], and angiogenesis [23]. Beyond these important functions however, the gut microbiota generates a complex network of metabolic pathways, as the bacterial gene set is approximately 150 times larger than the human. Therefore, one of the key functions of the gut microbiota is its enzymatic capability for enabling the acquisition of vitamins and bioactive compounds. For instance, through the fermentation of non-digestible substrates, like dietary fibers, anaerobic cecal and colonic bacteria produce short chain fatty acids (SCFA). SCFA are saturated aliphatic organic acids present mainly in the intestine, such as acetate, propionate, and butyrate [24].

The bidirectional crosstalk between the immune system and the microorganisms living in the gastrointestinal tract has established connections that extend beyond the metabolic nature. The high bacteria density present in the lower intestine is only separated from tissues and organs by a thin epithelial layer, which represents a great health challenge for the host. If commensal bacteria are not tightly controlled, a bacteria outgrowth could lead to inflammation and sepsis (commensals turn into pathobionts). Therefore, the host immune system is essential for preserving homeostasis with the microbial communities. The mucus layer, the secretion of antimicrobial peptides (AMP) and immunoglobulin A (IgA) by lamina propria plasma cells, and the presence of dendritic cells (DC) are some of the immune system mechanisms that help to stratify the luminal microbes, to minimize the interaction of the microorganisms and the immune system, and to reduce bacterial-epithelial contact [25] (Figure 1). Additionally, the immune system can modulate the gut microbiota composition at a species level [26,27,28]. The microbiota-host interaction works two ways and commensal microorganisms can shape mammalian immunity as well. Accumulating evidences show that gut flora impacts the immune system predominantly through small molecules (bacterial metabolites and other molecules) [29,30,31], which affects the balance between pro- and anti-inflammatory mechanisms [32]. For example, SCFA not only act as key metabolites for mammalian cell metabolism, but they also regulate host immunity. They are implicated in the following processes: they facilitate the extrathymic generation of regulatory T (T reg) cells [32], they regulate the function of the colonic T regs [33], and furthermore, they modulate the function of DCs, including cytokine production [34,35], through their receptor GPR43 [36]. The complete maturation of the host immune system is determined by host-specific commensals; exposure to a non-specific gut commensal or its microbe-associated molecular patterns (MAMPs) is not enough to induce intestinal immune maturation and fails to provide protection against enteric pathogens [37,38]. Mice raised under germ-free conditions have an altered gene-expression profile of the intestinal epithelial-cell layer [20], significantly smaller Peyer’s patches [39], a reduced number of CD4^+^ T cells [40], and a reduced number of IgA-producing plasma cells [41,42]. Several studies have shown the influence of commensal microbes on T-cell phenotype and function in the gut. For instance, the segmented filamentous bacterium (SFB) is sufficient to induce the appearance of T helper cell type 17 (Th17) CD4^+^ T cells in the lamina propria [43]. In mice, bacteria from the *Clostridiales* clusters IV, XIVa, and XVIII can direct T reg differentiation and *Bacteroides fragilis* can induce mucosal tolerance by defining the T reg lineage differentiation pathway [44,45] (Figure 1). Moreover, changes in microbiota density can alter host metabolism and the frequency of immune populations like lamina propria FoxP3^+^CD4^+^ T regs [46]. Beyond the effects on intestinal and local immune physiology, the gut microbiome has systemic effects [47,48]. For instance, peptidoglycan can prime systemic innate immunity by activating neutrophils [49]; polysaccharide (PSA) from *B. fragilis* can promote the increase in systemic T helper cell type 1 (Th1) CD4^+^ T cells [39]; and SCFAs can regulate extrathymic/peripheral T regs [32] and DC cytokine production and function [34,35] (Figure 1).

Commensal bacteria can prevent colonization by pathogens via multiple mechanisms, such as the production of microbial metabolites. *Lactobacillus* species improve the intestinal barrier function by modulating epithelial cell tight junction proteins. Clusters IV, XIVa, and XVIII of *Clostridia* (via TGF-β) and *Bacteroides fragilis* (via polysaccharide A (PSA)) promote the differentiation and expansion of regulatory T (T reg) cells and segmented filamentous bacteria (SFB) stimulate the induction of intestinal T helper cell type 17 (Th17) cells. Microbiota antigens are sampled via transepithelial dendrites of dendritic cells (DC) through M cells. These DC migrate to the mesenteric lymph node and there, they induce T-cell differentiation. Activated CD4^+^ T cells also prime B cells, and IgA^+^ secreting plasma cells then migrate to the lamina propria. Upon translocation into the circulation, bacterial metabolites from commensal microbes have multiple effects. Peptidoglycan can prime neutrophils and PSA can promote the increase in Th1 CD4^+^ T cells. SCFAs can regulate extrathymic/peripheral T regs, modulate leukocyte trafficking and DC cytokine production and function.

Gut microbiota has an additional important role in defining the TCR and BCR repertoire. It is known that there are circulating memory CD4^+^ T cells reactive against commensal bacterial lysates [50]. For instance, using a spontaneous uveitis mouse model, Horai and colleagues showed that the activation of retina-specific T cells was dependent on gut commensal microbiota [51]. Moreover, colonic T regs have a different TCR repertoire than peripheral T regs, implying a recognition of local colonic antigens [52]. Regarding the influence of the commensal microbiota on the BCR repertoire, ileum resident memory B cells expressing IgA and IgG have antigen-specificities for commensal flora [53].

Changes in the normal gut microbiota, called dysbiosis, have been reported to be an important factor in the development of many diseases. Dysbiosis contributes to autoimmune disorders, including arthritis and inflammatory bowel disease, allergic [54,55,56,57,58] and metabolic diseases, such as obesity and diabetes [59] (Figure 2). A recent study determined that 13% of global cancer incidence is due to microorganisms [60]. Moreover, some specific bacteria have been demonstrated to be involved in the process of initiation and progression of carcinogenesis at epithelial barriers [61,62], both locally and systemically. Locally, colorectal carcinogenesis and its progression [63,64] is directly associated with gut microbiota through several mechanisms. Systemically, several studies have confirmed that oncogenesis and tumor progression of breast [65,66], pancreas [67], and hepatocellular carcinoma [68] are influenced by gut microbiota. In addition, the human microbiota has been implicated in modulating the efficacy and toxicity of cancer therapy, including chemotherapy, radiotherapy, and, more recently, immunotherapy [35,69,70,71,72,73,74]. All these studies provide strong evidence for a close and complex interplay between the gut microbiome, tumor development, and anti-tumor immunotherapies as explained in the following sections.

## 2. Gut Microbiota in Hematological Malignancies

The changes of the gut microbiota composition have been assessed both in mouse models and in a clinical setting in adult and pediatric patients with hematological diseases. Most of these patients receive chemotherapy or immunotherapy and many of them additionally receive either antibiotic prophylaxis or other treatments that impact on the microbiota composition and diversity. This fact makes the study of the microbiota in hematological patients particularly challenging.

Ataxia-telangiectasia (A-T) is an autosomal recessive disorder associated with a high incidence of lymphoma. To study the role of intestinal bacteria in the penetrance of lymphoma, Yamamoto and coworkers used an A-T mouse model. This group studied the lymphoma incidence in different mouse colonies harboring different bacterial communities and found that *Lactobacillus johnsonii* was more abundant in the more cancer-resistant mouse colony and it could even reduce systemic inflammation and genotoxicity when administered orally to the more cancer-prone colony. Moreover, Ataxia-telangiectasia mutated (ATM) gene-deficient (Atm^-/-^) mice that were exposed to a more sterile environment lived longer and had a reduced lymphoma penetrance. This seemed to be associated with a reduced systemic inflammatory state (reduced basal leukocytes and cytokine-mediated inflammation) [75].

A study of adolescent/young adult Hodgkin lymphoma (AYAHL) survivors found that they had fewer early childhood fecal-oral exposures compared with healthy controls, which suggests reduced exposure to infections in these patients over childhood. Furthermore, suppressed Th1 activity and an increased T helper cell type 2 (Th2) response has been reported in AYAHL [76]. During childhood, the acquisition of gut microbiota diversity changes from a more immature Th2 to a Th1-orchestrated immune profile [77]. Both an increase in Th2 cytokines and IgE in AYAHL and a decrease in cytotoxic T cells and NK cells in Hodgkin lymphoma (HL) might suggest a failure to make this Th2-to-Th1 change in AYAHL. One reason could be that the observed decreased diversity of the gut microorganisms could impact the development of AYAHL [78,79]. Another study reported that AYAHL survivors seemed to have a reduced number of rare gut microorganisms compared to the unaffected twin controls [80]. As we mentioned at the beginning of this section, it is highly controversial whether this reduced microbial diversity is due to the environment as an initial risk factor, the lymphoma itself, or the treatment the patients received. However, it has been observed that gut microorganisms cause oxidative stress that can affect carcinogenesis and influence different pathways associated with lymphomagenesis [81,82,83,84,85]. Indeed, many pathogens have been directly associated with lymphomagenesis, such as Epstein-Barr virus (EBV), human herpesvirus 8 (HHV-8), human T-cell leukemia virus type 1 (HTLV-1), and *Helicobacter pylori* (HP), although the details regarding this are out of the scope of this review.

To study the influence of antibiotics that modulate intestinal microbiota in the efficacy of antineoplastic treatment, a group in Germany carried out a study on patients with relapsed lymphoma that were treated with cisplatin and patients with chronic lymphocytic leukemia (CLL) that were treated with cyclophosphamide from the CLL8 trial (NCT00281918) and the Cologne Cohort of Neutropenic Patients (NCT01821456). Among the 122 patients with relapsed lymphoma and the 800 patients with CLL, those treated with anti-Gram-positive antibiotics achieved a significantly lower overall response rate (ORR) and progressed significantly earlier [86]. The use of anti-Gram-positive antibiotics was independently associated with reduced progression-free survival (PFS) and overall survival (OS) in the multivariate analysis. A negative impact of anti-Gram-positive antibiotics on the efficacy of cyclophosphamide and cisplatin has also been observed in mouse models [74,87].

As mentioned before, commensal bacteria are involved in the differentiation of Th17 cells, which are characterized by their production of IL-17 and play a critical role in inflammation [88]. Calcinotto and colleagues studied the influence of intestinal microorganisms on multiple myeloma (MM) genesis in a mouse strain that develops a de novo disease mimicking MM. The commensal bacteria *Prevotella heparinolytica* was found to promote the differentiation of Th17 cells colonizing the gut; these cells migrated to the bone marrow (BM) where they favored tumor progression. *Prevotella heparinolytica* was also found to promote the progression of MM via an eosinophil-mediated inflammation [89]. A study in humans evaluated whether alterations in the intestinal flora are associated with relapse after allogeneic hematopoietic stem cell transplantation (allo-HSCT) and it was reported that microorganisms from the genus *Eubacterium* were associated with a reduced risk of MM relapse after allo-HSCT (as explained in more depth below) [90].

Decrease in bacterial diversity leads to a reduced colonization resistance against invading pathogens and has been associated with different pathological conditions such as inflammatory bowel disease, diabetes and obesity, as explained previously [91]. Several groups have reported the significant decrease in bacterial diversity both in acute lymphoblastic leukemia (ALL) and in acute myeloid leukemia (AML), and this effect has even been observed five years after the diagnosis [92,93,94,95,96,97,98].

In the following sections, we outline how the microbiota influences the response to treatment, including hematopoietic stem cell transplantation, both allogeneic and autologous.

## 3. The Treatment of Hematological Malignancies and Microbiota

Live wild-type bacteria can affect the efficacy of some anti-cancer agents either positively or negatively, in vitro and in vivo, most likely via enzymatic modifications (Table 1). Thus, in multiple reports, the efficacy of 20% of tested chemotherapy drugs was increased, the efficacy of 30% was decreased, and 50% were unaffected by bacteria. The cytotoxicities of cladribine, gemcitabine other commonly-used chemotherapy agents, like etoposide, and also anti-cancer antibiotics, such as doxorubicin, were decreased by bacteria. In contrast, bacteria increased fludarabine and 6-mercaptopurine cytotoxicity [99]. Moreover, these in vitro observations were replicated in an experimental mouse model where it was demonstrated that bacteria could hamper the effects of a selected drug, namely gemcitabine.

In addition to bacteria affecting drug effectiveness in patients with hematological malignancies, the microbiota is itself influenced by chemotherapeutics. Cyclophosphamide is commonly used in the treatment of patients with lymphoma, MM, and AL amyloidosis, as well as part of stem cell transplantation conditioning and chimeric antigen receptor (CAR) T cell lymphodepletive treatment [74]. These drugs can alter the composition of microbiota in the small intestine of mice, inducing the translocation of selected species of Gram-positive bacteria into secondary lymphoid organs. These bacteria stimulate the generation of a specific subset of “pathogenic” Th17 cells and memory Th1 immune responses. On the other hand, with platinum chemotherapy (used in some regimens for relapsed lymphoma, for example), infiltrating myeloid-derived cells responded poorly to therapy in germ-free mice, resulting in deficient production of reactive oxygen species and cytotoxicity after chemotherapy, as previously mentioned [87]. Thus, optimal responses to cancer therapy may require an intact commensal microbiota.

## 4. Allogeneic Stem Cell Transplantation

Allo-HSCT is a potentially curative modality for treating high-risk hematological malignancies (lymphomas and leukemias) and non-malignant conditions, such as aplastic anemia and inherited diseases. This procedure employs a conditioning regimen, including chemotherapy, radiation, radioimmunotherapy, and/or antibody-based immunotherapy with the goal being to reduce residual malignant cells, deplete the bone marrow hematopoietic cells, and suppress the patient’s immune system. Then, the patient receives an infusion of donor hematopoietic stem cells that reconstitutes hematopoiesis. The graft additionally contains allogeneic T-cells, which can attack residual malignant cells (graft vs. tumor effect) or healthy host tissues (graft vs. host disease (GvHD)) [100]. GvHD is still a leading cause of morbidity and mortality [101,102]. Acute GvHD (aGvHD) occurs in about 30–50% of patients and classically develops within 100 days after transplantation; however, late-onset aGvHD may develop even later [103]. Unfortunately, mortality remains high; the 2-year survival rate for patients with grade III and IV aGvHD is 25–30% and 1–2%, respectively [101]. Regarding pathophysiology, three phases have been described. Phase 1 is characterized by tissue damage due to a conditioning regiment that causes release of inflammatory cytokines and activation of host antigen-presenting cells (APC). During the second phase, host and donor APCs activate lymphocytes to produce Th1 cytokines and in phase 3, T cells migrate to target tissues and produce direct damage, especially in the skin, the gut and the liver [102,104]. Chronic GvHD (cGvHD) can occur with or without previous aGvHD and occurs in approximately 40% of patients [103]. The skin, the liver, the lungs and the gastrointestinal tract are the principal target organs [105]. The diagnosis of GvHD is predominantly based on clinical findings, supported by tissue biopsy in the case of aGvHD [106]. Strategies to reduce GvHD include optimization of HLA matching, T-cell depletion, and the use of immunosuppressive prophylaxis [107]. The first-line treatment for acute and chronic GvHD is based on corticosteroid therapy, however 35 to 50% of patients become refractory to this treatment. Responses to subsequent lines of therapy are also poor and there is no accepted standard of care treatment [108,109].

In recent years, the role of the intestinal microbiota in the development of infectious complications, GvHD, and mortality after allo-HSCT has been increasing. Allo-HSCT has been found to be related to a significant decrease in the diversity of the intestinal microbiota and it is believed to be due to the combination of various factors such as conditioning regimens, antibiotics, changes in diet, and intestinal inflammation [110,111,112]. About 50 years ago, different groups showed that, after receiving an allo-HCST, prolonged survival was observed in germ-free (gnotobiotic) animals and animals decontaminated with high-dose antibiotics compared with conventional mice due to reduced GvHD [113,114,115,116,117,118,119]. These results suggested that the microbiota plays a critical role in GvHD, and the absence of microbiota protects against it. However, these germ-free animals had an aberrant immunity and an anomalous development [120,121]. Efforts were made to bring these results into clinical practice. Diverse clinical trials and observational studies analyzed the effects of intestinal decontamination using broad spectrum antibiotics or lamina-airflow isolation rooms. However, the majority of these publications were contradictory and the outcomes were inconsistent [122,123,124,125,126,127,128,129,130]. In an attempt to narrow the antibiotic spectrum, the use of non-absorbable (rifaximin) or anti-anaerobic (metronidazole) antibiotics has been associated with a decrease in the rate and severity of GvHD [127,131,132,133]. Nevertheless, no standardized protocol for prophylactic antibiotics during allo-HSCT has been established [134].

The reduction in the diversity of the intestinal flora and domination by a single taxa at the time of neutrophil engraftment after allo-HSCT has been associated with reduced OS, increased transplant-related mortality, and GvHD-related mortality [135,136,137]. A recent publication showed that patterns of loss of diversity across different transplantation centers and geographic locations are similar. These findings highlighted that higher diversity of gut microbiota was associated with lower mortality, mainly in the subgroup of patients that received unmodified T-cell-repleted grafts [138]. These outcomes have been supported in a recent meta-analysis that highlights the importance of microbiota diversity and the drawbacks of intestinal decontamination [139].

One of the main limitations of the first studies carried out in this field is that most of them did not characterize the microbiota using high-throughput sequencing technology. Advanced molecular microbiological methods have elucidated that domination of a single taxa microorganism is associated with detrimental outcomes. For example, in mouse models the expansion of *Lactobacillus* and the depletion of Clostridiales were associated with worse GvHD [140]. Human studies have indicated that the presence of *Blautia* bacterial species is associated with reduced risk of GvHD-related mortality and increased OS [136]. This could suggest that certain microorganisms alleviate inflammation and could be used therapeutically [141]. However, a randomized clinical trial using the probiotic *Lactobacillus rhamnosus* GG did not show any benefit against GvHD after allo-HSCT [142]. Furthermore, lower levels of Parabacteroides and Bacteroides species were associated with aGvHD [143] and a greater abundance of *Firmicutes*, *Proteobacteria*, and Enterobacteriaceae was linked with increased mortality and GvHD [144], while a greater abundance of Lachnospiraceae and Actinomycetaceae was associated with better outcomes [135]. Similarly, monodomination of Enterococcus, *Escherichia coli* or *Prevotella* spp. was related to the presence and severity of GvHD [145,146,147,148]. Recently, an elegant study confirmed the role of fecal dominance by Enterococcus spp. in the development of aGvHD and for increased GvHD-related and overall mortality after allo-HSCT. They identified a microbiota-intrinsic mechanism dependent on lactose utilization that favors the expansion of *Enterococci*, whereby dietary lactose depletions attenuate the outgrowth of enterococci [149]. Contrastingly, Clostridiales have an important anti-inflammatory role, which includes upregulation of T regs, through the production of butyrate. Increased levels of butyrate aid the recovery of damaged intestinal epithelial cells after allo-HSCT and the depletion of anti-inflammatory *Clostridia* spp. precedes the development of GvHD [150]. Overall, there is sufficient data to confirm the role of the microbiota in immune reconstitution and immunosurveillance. In this context, Peled et al. found associations between the abundance of a bacterial group, mostly of *Eubacterium limosum*, and relapse after allo-HSCT [90]. However, the potential protective mechanism that could be provided by this bacterium is yet to be elucidated.

Regarding conditioning regimens, these can modify the composition of the intestinal microbiota by decreasing *Firmicutes* (including *Blautia* species), *Bifidobacteria*, and *Clostridium cluster XIV* and by increasing *Enterococcus* and *Proteobacteria* (including *Escherichia* species). However, it has not been possible to demonstrate a causal relationship between conditioning regimens and the diversity of intestinal bacteria due to other confounding factors such as the concurrent use of prophylactic antibiotics [151,152]. A very recent report from Memorial Sloan Kettering Cancer Center (MSKCC) described the association between neutrophil, lymphocyte, and monocyte populations during hematological recovery and the microbiota dynamics in hundreds of patients who received stem cell transplantation, with positive associations between both obligate anaerobe fermenters and *Staphylococcus* and immune cell dynamics [153]. The role of microbial metabolites has recently been associated with the development of GvHD [134]. Indole and indole derivates limited intestinal epithelial damage and reduced GvHD while preserving graft versus tumor activity, and this was related to the upregulation of genes associated with type I interferon responses [154]. A low urinary level of indoxyl sulfate (a metabolite of tryptophan) has been associated with the development of GvHD; therefore, it could be a potential biomarker of a disrupted microbiome, as well as risk of GvHD [155]. Low levels of propionate have been detected in stool samples from patients with GvHD [143]. In murine intestinal tissue, reduced levels of butyrate were found after allo-HSCT, and the administration of butyrate improved the integrity of the intestinal epithelial cells and reduced the severity of GvHD [156]. All these data suggest that intestinal microbiome-derived metabolites may modulate intestinal damage and mitigate GvHD [156] (Table 2).

## 5. Autologous Stem Cell Transplant

Autologous hematopoietic stem cell transplantation (ASCT) is a multistep procedure originally developed for the treatment of hematological malignancies. This is part of the first-line treatment for MM, and an alternative treatment modality for lymphomas and a few solid tumors. After administration of G-CSF and/or chemotherapy, hematopoietic stem cells are harvested. The infusion of these cells is used to bridge hematopoietic failure after high-dose chemotherapy, usually melphalan for MM and combination regimens for lymphoma, such as BEAM (BCNU, etoposide, cytarabine, and melphalan). In this sense, autologous stem cell support is not a “transplant”; however, the term “ASCT” is commonly used. More than half of the stem cell transplants performed in Europe are autologous, most of them for lymphoid malignancies (plasma-cell disorders (mainly MM), non-Hodgkin lymphoma, and Hodgkin lymphoma), although the procedure has been also adapted for the treatment of severe immune-mediated disorders [165,166,167,168]. The source of stem cells for 99% of all autologous transplant procedures is peripheral blood.

There is limited information about the impact of microbiota on ASCT. Preliminary results showed significant changes in the oral microbiome of 51 patients after ASCT, which returned to its pre-chemotherapy composition after three months. However, changes in microbial diversity were more pronounced and rapid in patients who developed mucositis compared to patients who did not [161]. For non-Hodgkin lymphoma, ASCT caused a decrease in *Firmicutes* and *Actinobacteria* abundance but an increase in *Proteobacteria* abundance in fecal samples collected after exposure to conditioning compared to baseline fecal samples [169]. A small pilot study with 15 patients with plasma-cell dyscrasia showed that microbiome composition present at baseline was associated with the incidence and severity of post-transplantation nausea, vomiting, and culture-negative neutropenic fever, as well as with the rate of neutrophil engraftment [170]. Other studies have included ASCT in the design, but the majority of them received allo-HSCT [171]. There are many aspects still to explore regarding the relationship between the microbiome and ASCT, and the association of the microbiota with other immune complications such as engraftment syndrome [172] would be of clinical interest for further studies (Table 2).

## 6. Antibiotics and Acute Myeloid Leukemia

The impact of microbiota composition has not been studied in detail in patients receiving induction therapy for AML. However, antibiotic prophylaxis during chemotherapy could be a factor to consider when evaluating the microbiome in these patients. A report of 60 patients, most with AML (42%) or MM (37%), showed that the treatment of neutropenic fever with beta-lactam antibiotics reduced the diversity of the gut microbiome in comparison to when the prophylactic levofloxacin was used [173]. Moreover, another study reported that baseline microbiome diversity was a strong independent predictor of infection during induction chemotherapy. Higher baseline levels of *Porphyromonadaceae* appeared protective against infection, while carbapenem use was associated with deeper changes in the microbiome and with infection susceptibility [161]. This microbiome-based information could be useful to design interventional strategies and optimize antibiotic administration in clinical practice for patients with AML.

## 7. First Line Treatment in Multiple Myeloma

Survival trends for patients with MM appear to be improving in the last decade, and these better outcomes are related to the use of novel therapeutic agents. The introduction of ASCT, followed by immunomodulatory drugs (thalidomide, lenalidomide, pomalidomide) proteasome inhibitors (bortezomib, carfilzomib, ixazomib), and monoclonal antibodies (daratumumab), has improved the OS in older patients and lowered early mortality rates [174,175,176]. A very recent report from MSKCC [163] of 34 MM patients who completed first-line therapy and were candidates for lenalidomide maintenance sought to define the connection between the patient intestinal microbiome and MM disease progression. Patients were evaluated for minimal residual disease (MRD) status after completion of induction and before the start of maintenance therapy and stool samples were obtained for DNA sequencing to identify the microbiome composition. A higher relative abundance of the butyrate producer *E. hallii* was observed in patients without MRD in their BM when compared to those that were MRD^+^. *F prausnitzii* was identified as another microbe potentially associated with negative MRD status after initial therapy. Future studies are needed to confirm these findings about intestinal microbiota composition and deeper treatment response. In general, information about the link between myeloma treatment and the microbiome is scarce.

## 8. CAR T Cells

CARs are synthetic fusion proteins that redirect lymphocytes to recognize and eliminate cells that express a target antigen on its surface. A CAR is endowed with four fundamental components: an extracellular antigen binding domain or single-chain variable fragment (scFv) derived typically from the immunoglobulin structure; a spacer or hinge region; a transmembrane domain from CD8α or CD28 molecules; and the intracellular or activation domains derived from the CD3ζ subunit of the TCR and one or two co-stimulatory domains derived from CD28 or 4-1BB molecules, among others. Synthetic engineering of T-cells expressing CARs against CD19 antigen have shown outstanding results against B-cell malignancies in clinical trials. Therefore, the US Food and Drug Administration (FDA) and the European Medicines Agency (EMA) have approved Tisagenlecleucel (Kymriah) for relapsed/refractory (R/R) B-cell acute lymphoblastic leukemia (ALL) and R/R diffuse large B-cell lymphoma, and Axicabtagene Ciloleucel (Yescarta) for R/R diffuse large B-cell lymphoma as well. Recently, the FDA has also approved brexucabtagene autoleucel (Tecartus) for R/R mantle cell lymphoma [177,178,179,180,181]. The outstanding results of anti-CD19 CAR T cells have led to numerous clinical trials being carried out on diverse hematological and solid cancers. In this sense, several encouraging results have been presented using CAR T-cells/CART-cell therapy targeting BCMA in MM patients. This therapy has emerged as a potential treatment strategy for R/R MM patients, and it is expected that in the next few months the first anti-BCMA CAR will be approved [182,183].

Fecal samples from 25 patients at MSKCC were collected pre-CAR T cell infusion [164] and microbiota composition was profiled by 16S sequencing to determine the correlation between microbiome composition and the efficacy of CAR T cell therapy. The patients were adult recipients of CAR T cells who varied regarding conditioning regimen, CAR construct, and underlying diagnosis, which included hematologic and solid malignancies. The study found an increased representation of bacterial taxa in the microbiome of patients who achieved a complete remission (CR) versus those who did not. *Oscillospi*raceae, Ruminococcacaeae, and Lachnospiraceae were enriched in patients with CR and Peptostreptococcaceae was more abundant in patients who did not achieve a CR. A higher abundance of Lachnospiraceae was found in those who experienced some toxicity (cytokine release syndrome or neurotoxicity of grade 1 to 4), while Peptostreptococcaceae was more abundant in patients who did not have toxicity. Although very preliminary, these data indicate that features of the microbiota may correlate with outcomes of CAR T cell therapy (Table 2).

## 9. Outlook-Future Perspective

Gut microbiota has been studied in detail for the last decade and overall data suggest the great impact of intestinal microorganisms upon carcinogenesis, cancer proliferation, HSCT outcome, and response to anticancer therapeutics. Hence, gut microbiota modulation is an exciting and important field of research that will likely be used as a complement to existing therapies, either to enhance the efficacy of treatments or to diminish first-line treatment side effects. To date, several microbiota-targeted approaches have been used to modulate the gastrointestinal bacterial composition (Figure 2).

Dietary intervention can change the gut microbiota composition within 24–48 h at the level of species and family, but not phyla [184]. In addition, the circadian rhythm needs to be accounted for, as at least 10% of Operational Taxonomic Units (OTUs) exhibit diurnal oscillations in their abundance [185]. A Western-style diet, which is known to be linked to higher levels of inflammatory markers and with inflammation-related chronic diseases [186], has been associated with CLL. The author of the study hypothesized that the Western diet microbiota could lack the diversity to establish a balanced immune response, therefore suggesting that a percentage of CLL cases could be prevented by a change in dietary patterns [187]. Dietary intervention is also being tested in a clinical trial of subjects undergoing allo-HSCT in which a potato-based resistant starch is administered for 100 days, beginning just before the conditioning phase. The aim is to increase the butyrate level inside the intestine to mitigate GvHD [188] (Table 3).

Prebiotics, like oligosaccharides, are components that can promote the growth and function of beneficial microorganisms. They can have a direct effect inhibiting pathogen colonization [189,190]. There are a few clinical trials studying the effects of prebiotics on GvHD. A pilot phase I trial, now completed but without published results, tested the side effects and best dose of fructo-oligosaccharides in patients with hematological malignancies undergoing HSCT, with the main goal being to reduce the incidence of GvHD [188,191]. A phase I/II clinical trial, not recruiting yet, has been approved to determine whether a prebiotic, galacto-oligosaccharide, a host carbohydrate, can modulate the microbiome to prevent GvHD after allo-HSCT. During phase I, the maximum tolerated dose will be determined and in phase II, participants will be randomized to receive the prebiotic or a placebo [192]. Lysozyme is an enzyme that plays a role in defense against gastrointestinal pathogens. It is effective against Gram-positive and Gram-negative bacteria and promotes beneficial microbes and reduces detrimental microbes within gut microbial communities [193,194]. A phase I trial will study the safety and efficacy of goat milk that is genetically engineered to produce human lysozyme in the context of preventing GvHD in patients with hematological malignancies undergoing HSCT [195]. To date, the use of prebiotics as an adjuvant treatment in the hematological malignancies setting is hindered by a lack of solid data. However, great efforts are being made in this field and the results of different ongoing clinical trials are eagerly awaited (Table 3).

Probiotics are beneficial microorganisms that, in the correct amount, confer health benefits for the host. Besides restoring microbial dysbiosis and preventing colonization of pathogenic bacteria [196], probiotic byproducts can lower the intestinal infection risk and inflammation [197]. Probiotics can be directly introduced as defined microbial strains, engineered microbes, or by FMT. As a defined strain strategy, there is a randomized open label pilot study testing CBM588 (*Clostridium butyricum* CBM 588 Probiotic Strain). The aims of this study are the following: (1) to assess the side effects of this probiotic, and (2) to evaluate if it improves the clinical outcome in patients undergoing allo-HSCT by increasing gut microbiota diversity [198]. FMT is an effective approach to restore a dysbiotic intestinal community, therefore returning the homeostasis with the microbial communities. This is achieved by administering a fecal solution from a healthy donor into the gastrointestinal tract of a recipient. Healthy donors are selected after a thorough screening to discard harmful pathogens or family history of autoimmune, metabolic or malignant diseases. FMT can be delivered via colonoscopy, enema, nasogastric or nasojejunal tube and gastroscopy. Moreover, FMT can now be performed orally through a freeze-dried capsule containing the fecal matter. This therapeutic procedure represents an innovation in the hematological field with a remarkable potential to minimize the side effects of standard treatments like antibiotics or chemotherapy. Currently, FMT is only approved for the *Clostridioides difficile* infection (CDI) but there are already some promising results. Refractory immune checkpoint inhibitor-associated colitis was successfully treated with FMT in two patients [199]. At this time, over 10 clinical trials are being run to establish if FMT is a viable treatment for GvHD [200,201,202,203,204,205,206,207,208], as well as for gastrointestinal complications in AML patients [209,210], none with posted results. Importantly, some of the clinical trials are already testing FMT as a first-line treatment for severe aGvDH [201,208] or as an upfront treatment to avoid or reduce the incidence of GvHD [209,210] or multidrug-resistant bacteria (MDRB) [210] in HSCT patients. Although it has been shown that in the recurrent CDI prevention context, FMT administered orally via capsules is as effective as administered by colonoscopy [211], these clinical trials are testing several FMT administration options: oral administration through a freeze-dried capsule [198,200,202,203,206,207,209], colonoscopy/gastroscopy [201,205,208], rectal enema [210], and nasojejunal tube/gastroscopy [204] administration. While an increasing number of studies are testing FMT as a treatment, it has encountered several difficulties that have halted its application in the clinic. A patient died from a drug-resistant *Escherichia coli* bacteremia transmitted by FMT after an allo-HSCT [212]. Moreover, the FDA issued a safety alert in 2020 after the death of patients receiving FMT for CDI. Patients developed infections caused by enteropathogenic *Escherichia coli* (EPEC) and Shiga toxin-producing *Escherichia coli* (STEC) [213]. Hence, a better stool donor screening technique to avoid the transmission of harmful microorganisms is required and multiple clinical trials are ongoing to confirm the safety and viability of FMT (Table 3).

Postbiotics are bacterial bioactive compounds that can be delivered directly to a patient without targeting the gut microbiota composition. This is done by the exogenous administration of a specific metabolite, like SCFAs, flavonoids, the organic acid taurine, and indole derivatives. A recent study showed that higher circulating concentrations of butyrate and propionate are associated with protection from cGvHD in allo-HSCT patients [214]. On the other hand, the gut microbial metabolite trimethylamine N-oxide (TMAO) aggravates GvHD in mice, enhancing M1 macrophage polarization via NLRP3 inflammasome activation [215]. These two examples highlight the therapeutic potential of postbiotics in hematological malignancies (Table 3).

Antibiotic strategies, that include selecting more narrow-spectrum agents, are currently being tested in clinical trials in the HSCT setting in order to preserve intestinal microbiota composition and therefore reduce the incidence and severity of aGvHD [216] (Table 3).

## 10. Concluding Remarks

Gut microbiota can shape the immune system well beyond the gastrointestinal tract and this is key to comprehending its role in health and disease. Understanding this interplay will help us to identify novel targets in the design of new approaches to treat hematological malignancies. For instance, a possible approach to treat cancer as an adoptive T-cell therapy (ACT), like CAR-T therapy, would be the use of cross-reactive T cells. Cross-reactive T cells could recognize bacterial antigens but are able to recognize tumor-associated antigens as well. Migrating bacterial antigen-loaded DCs would travel to secondary lymphoid organs and potentially prime cross-reactive anti-tumor T cells. These T cells could be potentially used as an ACT themselves or as an adjuvant for CAR-T cell therapy. Furthermore, gastrointestinal flora modulation is a promising approach to improve the efficacy of hematological malignancies therapies, especially to avoid the serious complications related to those treatments. The clinical trials ongoing in this field will help to establish this approach as a therapeutic tool, adding a valuable personalized medicine resource in the standard care treatment of hematological malignancies (Figure 2). FMT is the most advanced option to move forward. As outlined in this review, multiple clinical trials are being developed in the setting of hematological malignancies. Several key factors must be taken into consideration, like the source of the fecal transplant, autologous vs. healthy donor, its way of administration, the dose, and the timing of infusion. It is also important to acknowledge that different hematological malignancies, or their complications, may need different approaches. Moreover, once FMT is validated as a therapeutic tool beyond CDI, it will be necessary to establish the required technology and to standardize the donor microbiota. As mentioned before, besides FMT, other strategies are being studied in pre-clinical models and clinical trials. A lot of resources are being used to uncover the most efficient way to modulate the gastrointestinal flora, but to date, the use of dietary intervention, prebiotics, probiotics, or postbiotics is hindered by a lack of solid and reproducible data. This highlights the need for more basic and preclinical research to better understand the mechanisms of action behind gut microbiota modulation.

## Figures and Tables

**Figure 1 ijms-22-01026-f001:**
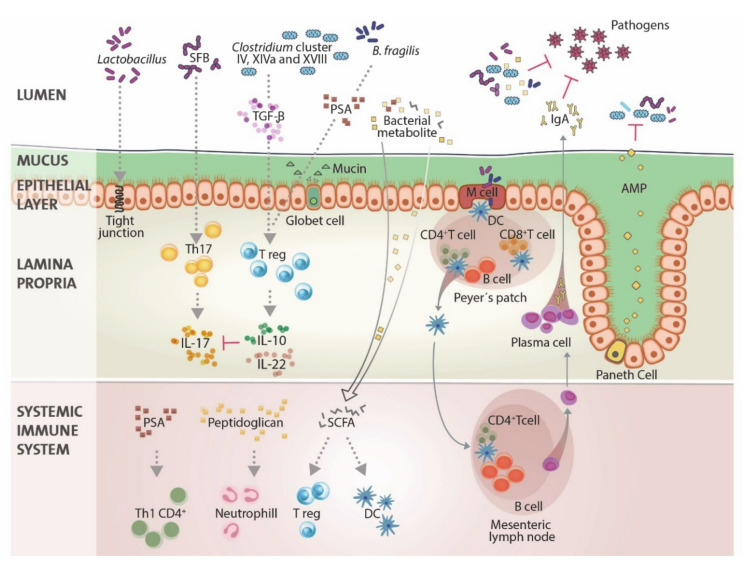
Dynamic interactions between the gut microbiota and the immune system.

**Figure 2 ijms-22-01026-f002:**
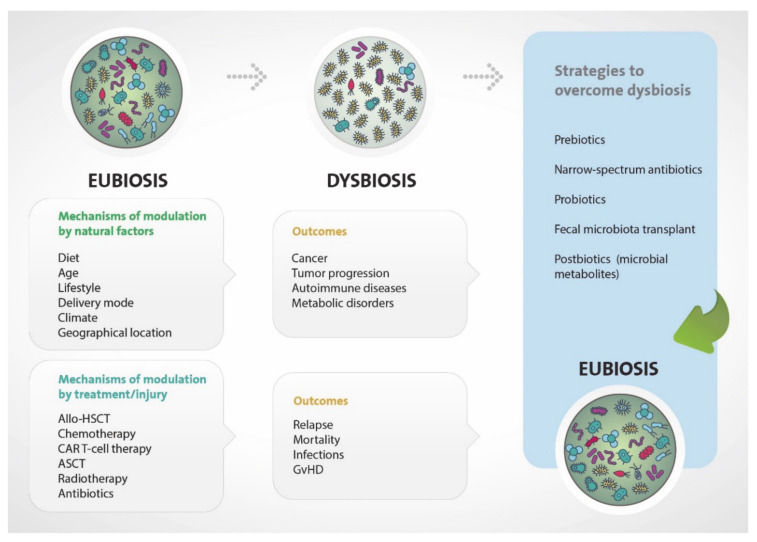
Mechanisms of gut microbiota modulation, consequences, and potential solutions.

**Table 1 ijms-22-01026-t001:** Summary of the potential effects of gut microbiota on anticancer agents used in clinical hematology.

Mechanism	Outcome	Involved Drug (s)
Enzymatic modification of the drugs	Decreased levels	Cladribine, gemcitabine, doxorubicin, Idarubicin, etoposide, mitoxantrone [99]
	Increased activity	Fludarabine, 6-Mercaptopurine [99]
Translocation of Gram-positive bacteria	Generation of Th17 and Th1 lymphocytes	Cyclophosphamide [74]
Modulation of genotoxicity	Reduction of DNA damage and apoptosis	Oxaliplatin, cisplatin [87]

**Table 2 ijms-22-01026-t002:** Summary of outcomes induced by type of treatment and microbiota disruption in humans.

Treatment	Microbiota Feature	Outcome
Allogeneic hematopoietic stem cell transplantation	Decontamination of gut anaerobes	Lower risk of GvHD [127,132]
	Decreased duodenal Paneth cells	Higher GI GvHD and NRM [148]
	Low intestinal microbiota diversity	Higher TRM, lower OS and GvHD-related mortality [135,138]
	*Enterococcus* spp. domination	Increased GI GvHD severity [145] Increased GvHD-related and overall mortality [146,157] Associated with blood stream infections [110,158]
	*Blautia* abundance	Reduced GvHD-related mortality [136]
	*Clostridia* spp. depletion	Increased GvHD [150]
	*Barnesiella* spp. abundance	Protection against *Enterococcus* domination [159]
	*Akkermansia muciniphila* domination	Mucus degradation [128]
	Lactobacillales domination	Associated with GvHD development [140]
	*Eubacterium limosum* abundance	Lower risk of relapse or progression, higher OS [90]
	Picobirnivirus presence	Severe GI GvHD [160]
	Lower urinary 3-indoxyl sulfate	Higher risk of GvHD, higher TRM, lower OS, higher dysbiosis [145,155]
Autologous stem cell transplantation	Reduction in diversity index and an increased dominance index	Development of mucositis [161]
	Decreased in Firmicutes and Actinobacteria and increased in Proteobacteria	Development of GI mucositis [161]
Chemotherapy	Baseline levels of Porphyromonadaceae	Predictor of infection during induction for acute myeloid leukemia [162]
	Relative abundance of *E hallii*	Higher negative minimal residual disease rate in bone marrow for multiple myeloma [163]
CAR T-cell therapy	Oscillospiraceae, Ruminococcacaeae and Lachnospiraceae enriched	Association with complete remission after CAR T cell therapy [164]
	Higher abundance of Lachnospiraceae	Development of cytokine release syndrome and/or neurotoxicity [164]

GI gastrointestinal; GvHD graft versus host disease; NRM no-relapse mortality; OS overall survival; TRM, transplant-related mortality.

**Table 3 ijms-22-01026-t003:** Interventional clinical trials employing microbial products of hematological malignancies treatment and its complications.

Condition	Intervention	Administration	Primary Aim	Phase	Participants	Status	Identifier
aGvHD	FMT	Orally via capsule	Efficacy, safety, and tolerability	I	10	NYR	NCT04280471
GI aGvHD	FMT	Colonoscopy or gastroscopy	Efficacy and safety	II	30	R	NCT03812705
GI aGvHD	FMT	Nasojejunal tube	Efficacy and safety	I	15	NYR	NCT03549676
GI aGvHD	FMT	Orally via capsule	Efficacy and safety	I/II	20	R	NCT04269850
GI aGvHD	FMT	Orally via capsule	Efficacy and safety	II	17	NYR	NCT04059757
GI aGvHD	FMT	Colonoscopy or gastroscopy	Efficacy	III	15	R	NCT03819803
GI aGvHD	FMT	Orally via capsule	Efficacy and safety	N/A	10	R	NCT04622475
aGvHD	FMT	Orally via capsule	Feasibility and efficacy	I	11	NYR	NCT04139577
GI aGvHD	FMT	Colonoscopy or duodenal nutrition tube injection	Efficacy	I	30	R	NCT04285424
Allo-HSCT, AML	FMT	Orally via capsule	Efficacy and incidence of infections	II	120	R	NCT03678493
AML	FMT	Rectal enema	Evaluation in dysbiosis correction and in MDRB eradication	I/II	20	C	NCT02928523
aGvHD	FMT	Not specified	Efficacy	II	24	C	NCT03359980
Allo-HSCT	FMT	Orally via capsule	Feasibility and efficacy	I	18	C	NCT02733744
Allo-HSCT	Potato-based dietary starch	Orally	Incidence of grade II-IV GVHD	II	70	R	NCT02763033
Allo-HSCT	Fructo-oligosaccharides prebiotic	Orally	Safety and tolerability	I	15	C	NCT02805075
Allo-HSCT	Galacto-oligosaccharide prebiotic	Orally	Tolerability and incidence of Grade II-IV aGVHD	I/II	128	NYR	NCT04373057
Allo-HSCT	Human lysozyme goat milk prebiotic	Orally	Safety and tolerability	I	36	NYR	NCT04177004
Allo-HSCT	*Clostridium butyricum* CBM 588 Probiotic Strain	Orally	Safety	I	36	R	NCT03922035

AML acute myeloid leukemia; Allo-HSCT allogeneic hematopoietic stem cell transplant; FMT fecal microbiota transplant; GI gastrointestinal; GvHD graft versus host disease; HSCT hematopoietic stem cell transplant; MDRB multidrug resistant bacteria; NYR, Not yet recruiting; R recruiting; C completed.

## Data Availability

Not applicable.

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
