# Peer review of "Gut Microbiota Influence in Hematological Malignancies: From Genesis to Cure"

_ijms, 2021, doi:10.3390/ijms22031026_

Round 1
Reviewer 1 Report
The text is very interesting and detailed on the most recent literature on intestinal microbiota and haematological diseases. In them, the literature concerning biology is reviewed together with the most relevant literature in the haematological field (allogeneic transplant, autologous transplant, AML and MM, CAR-T).
Table 1 summarizes the main causes that alter the gut microbiota, while Table 2 summarizes the studies that use microbiota modifiers to combat certain conditions such as GvHD.
Within all the paragraphs described, it would be worth dedicating one to FMT which represents an innovation in the medical field, capable of restoring intestinal microbiological diversity altered by antibiotic / chemo-radiotherapy treatments. The authors also describe how the cytotoxicity of some chemotherapeutic agents can be altered by some bacterial species. it would be interesting to summarize these findings in a table.
Author Response
Dear Reviewer,
We thank you for your positive comments and careful evaluation, which helped improve the manuscript. In response to your suggestions, we added a paragraph about Fecal microbiota transplant (lines 546-555) and we summarized the effects of gut microbiota upon chemotherapy agents cytotoxicity in a table (new Table 1, after section 3. The treatment of hematological malignancies and microbiota).
Kind regards
Reviewer 2 Report
The manuscript “Gut microbiota influence in hematological malignancies: from genesis to cure” is a relatively-well written review on the role of the human microbiota in disease, with emphasis on hematological malignancies. The manuscript thoroughly reviews the relatively limited but increasing data on the role of gut microbiota in several disease states, and on what we call dysbiosis. I especially liked tables 1 and 2 that effectively summarize the microbiota disruption by various treatments used in hematological malignancies (table 1) and the various interventional clinical trials employing microbial products (for the most part fecal microbiota transplants) for the treatment of GVHD and other conditions linked with hematological malignancies.
My main criticism is that the manuscript needs minor to moderate revisions from a native English speaker. More specifically, there are several typos. For example, in page 2, the authors write that antipsychotics and calcium-channel blockers can also affect de human microbiome. In the same page, they write: It has now been stablished, etc. Also, the last paragraph of page 3 and the first paragraph of page 4 have several mistakes (they the induced T-cell differentiation). Hence, less than though editing is apparent throughout the manuscript. Moreover, some sentences are strangely worded, to the point that it is relatively difficult to follow. For example, in page 1, the last sentence of the first paragraph of the introduction (The highly diverse bacterial community….) is slightly obscure, and although I fully understand the meaning, the sentence certainly needs editing to improve its readability.
Author Response
Dear Reviewer,
We thank you for your valuable comments. As suggested, the manuscript has been carefully revised by a native English speaker, which has improved our manuscript substantially.
Kind regards